# Microglial Activation: Key Players in Sepsis-Associated Encephalopathy

**DOI:** 10.3390/brainsci13101453

**Published:** 2023-10-12

**Authors:** Jiyun Hu, Shucai Xie, Haisong Zhang, Xinrun Wang, Binbin Meng, Lina Zhang

**Affiliations:** 1Department of Critical Care Medicine, Xiangya Hospital, Central South University, Changsha 410008, China; 218111146@csu.edu.cn (J.H.); xieshucai1990@foxmail.com (S.X.); 218102101@csu.edu.cn (H.Z.); wangxr1893@163.com (X.W.); mbb18154698607@163.com (B.M.); 2Hunan Provincial Clinical Research Center for Critical Care Medicine, Xiangya Hospital, Central South University, Changsha 410008, China; 3National Clinical Research Center for Geriatric Disorders, Xiangya Hospital, Central South University, Changsha 410008, China

**Keywords:** sepsis-associated encephalopathy, microglial activation, M1/M2 microglial polarization, cognitive impairment, therapeutic strategies

## Abstract

Sepsis-associated encephalopathy (SAE) is a common brain dysfunction, which results in severe cognitive and neurological sequelae and an increased mortality rate in patients with sepsis. Depending on the stimulus, microglia (resident macrophages in the brain that are involved in SAE pathology and physiology) can adopt two polarization states (M1/M2), corresponding to altered microglial morphology, gene expression, and function. We systematically described the pathogenesis, morphology, function, and phenotype of microglial activation in SAE and demonstrated that microglia are closely related to SAE occurrence and development, and concomitant cognitive impairment. Finally, some potential therapeutic approaches that can prime microglia and neuroinflammation toward the beneficial restorative microglial phenotype in SAE were outlined.

## 1. Introduction

Sepsis is a common cause of death among critically ill patients in intensive care units. According to the latest global estimates of sepsis incidence and mortality, 49 million people suffer from sepsis annually, resulting in 11 million deaths, comprising 20% of all deaths worldwide [1,2]. Sepsis is now defined as infection with organ dysfunction as determined using the Sequential Organ Failure Assessment (SOFA) score [3], and it was designated as a global health priority in 2017 by the World Health Assembly and the World Health Organization (WHO) [4]. A common complication of sepsis is diffuse brain dysfunction and cognitive impairment caused by infection outside the central nervous system (CNS), known as sepsis-associated encephalopathy (SAE) [5]. It frequently occurs in the absence of an overt infection of the CNS and manifests as a mild disturbance of consciousness, disorientation, cognitive impairment, convulsion, or deep coma [6].

While the pathogenesis of SAE is likely multifactorial and has not been fully elucidated, treating and diagnosing SAE are also equally challenging tasks. During clinical practice, the detection of abnormalities in electroencephalogram recordings and abnormal mental statuses, along with a clinical history, physical examination, laboratory tests, and neuroimaging evaluation, is carried out to diagnose SAE [7]. The etiology and pathogenesis of SAE are complex, and include microglial activation, blood–brain barrier (BBB) disruption, leukocyte infiltration, metabolic adaptations to systemic inflammation, bioenergetic shifts, cerebral coagulopathy or ischemia, oxidative stress due to inflammation, and mitochondrial dysfunction [8,9]. A localized and significant increase in CD68-positive microglia is observed in the brains of patients who die from septic shock due to severe systemic inflammation and increased microglial activity in the putamen, hippocampus, and cerebellum of the brain [10,11]. Additionally, the activation of pro-inflammatory microglia in white matter is observed in patients with sepsis; however, this is not as evident in gray matter. In contrast to brain inflammatory or ischemic diseases, the anti-inflammatory microglia markers CD163 and CD206 are not expressed in acute sepsis [12]. Microglial activation can cause neuronal injury or apoptosis via the release of inflammatory mediators, reactive oxygen species, neurotransmitters, and other substances. Microglia also secrete cytokines and chemokines that protect the brain from inflammatory responses. However, because of the regulation of white blood cell migration and neuronal repair, the long-term activation of microglia has minimal protective effects on neurons and worsens the inflammatory response in the brain [13]. In this review, we systematically searched common English databases, including PubMed, Web of Science, MEDLINE, and Embase, to investigate the critical role of microglia in SAE, and summarized the prospects of therapies targeting microglial activation and neuroinflammation to alleviate cognitive impairment in SAE in recent years.

## 2. Microglia in Homeostasis

Brain development and CNS homeostasis are normally regulated via microglia processes, which include programmed cell death, clearing apoptotic newborn neurons, and pruning axons and synapses that are developing. Throughout development and adulthood, microglial processes are highly mobile, continuously surveilling their local environment, making contact with neurons, axons, and dendritic spines [14]. Microglia modulate synaptic transmissions and facilitate neural circuit formation by devouring eliminated synapses in a complement-dependent manner [15,16]. Microglia are derived from yolk-sac-derived progenitors and constitute approximately 10% of brain cells and approximately 20% of all glial cells, which as the primary cleaners of the brain, engage in phagocytosis to eliminate dead neurons and minimize the accumulation of debris [17]. They are the macrophage-like myeloid innate immune cells of the brain and spinal cord, act as the main immune defense in the CNS, and are rapidly activated in most neurological diseases, including traumatic brain injury, stroke, Alzheimer’s disease, Parkinson’s disease, multiple sclerosis, schizophrenia, etc. [18,19].

The CNS has traditionally been considered immune-privileged owing to the BBB tight junctions between endothelial cells, the basal lamina of these endothelial cells, and astrocytic end-feet processes, which significantly reduce the infiltration of macromolecules and immune cells into the parenchyma. In addition, the brain lacks professional antigen-presenting cells and expresses low levels of major histocompatibility complex class I and II molecules [20]. Microglial activation can occur even with minimal disturbance, maintaining the homeostasis of the local brain parenchyma. Under physiological conditions, microglial processes are motile and exhibit a ramified morphology in the brain of healthy adults. However, microglia become activated and transform into a hypertrophic or amoeboid shape in neurodegenerative diseases and upon neuronal injury [21,22]. Similar to macrophages, microglia respond to invading pathogens by sequestering and inoculating microbes and limiting the effects of cell damage and necrosis [23,24]. These acute responses include migration, proliferation, phagocytosis, antigen presentation, and the release of various effector substances, including superoxide, nitric oxide, proteases, and anti-inflammatory (such as interleukin [IL]-10 and IL-4) and pro-inflammatory (such as IL-1β and IL-6) cytokines [25,26]. In addition to that, it has been demonstrated that microglial activation could damage the BBB through the release of MMP-2/-9 [27].

## 3. M1/M2 Microglial Polarization

The activation of microglia is a complex phenomenon characterized by a series of temporally, physiologically, and spatially regulated events that contribute to the observed morphological and functional alterations in these reactive cells. Recent research has shown that microglia are most diverse in developing, aged, and injured brains via the mapping of single cells of microglia in mice at various stages of development and after brain injury [28]. Depending on the milieu and factors that stimulate them, microglia can participate in classical activation, alternative activation, or acquired deactivation. Under physiological conditions, microglia are designated as “resting” while the reactive morphology is termed “activated” (with a rounder cell body, and fewer and shorter processes or an amoeboid-shaped cell) [29]. To model this change, typical experiments involve the exposure of microglial cells in vitro to stimuli such as apoptotic cells, lipopolysaccharide, inflammatory cytokines, or aggregated proteins [30]. Microglia can be phenotypically polarized to develop either a classical (proinflammatory, M1) or an alternative (anti-inflammatory and pro-healing, M2) phenotype (Figure 1). It should be noticed that this has only been demonstrated under experimental conditions, whereas microglia within the organism can adopt any phenotype from the spectrum between M1 and M2 phenotypes. During the progression of neuroinflammatory diseases, the balance between the M1 and M2 states of microglia is dynamic. M1 microglia dominate the injury site at the end stage of the disease when the immune resolution and repair processes of M2 microglia are impaired [31]. M1 microglia produce cytokines and chemokines (IL-1β, IL-6, IL-12, tumor necrosis factor α [TNF-α], and chemokine (C-C motif) ligand 2), express nicotinamide adenine dinucleotide phosphate oxidase, and generate reactive oxygen and nitrogen species.

Moreover, M1 microglia express histocompatibility complex II, CD11b, CD11c integrins, CD36, CD45, and CD47 costimulatory molecules. M2 microglia are capable of releasing several anti-inflammatory cytokines (IL-10; transforming growth factor β [TGF-β]), growth factors (insulin-like growth factor, fibroblast growth factor, and colony-stimulating factor 1), and neurotrophic growth factors (nerve-derived growth factor, brain-derived neurotrophic factor, neurotrophins, and glial cell-derived neurotrophic factor) [32,33,34].

However, some scholars have different opinions on the dichotomic rigid categories of M1/M2, which is inconsistent with the wide repertoire of microglial states and functions in development, plasticity, aging, and diseases which were elucidated in recent years [35]. The advent of single-cell technologies has provided clear evidence that microglia in the living brain do not polarize to either of these phenotypes, often co-expressing M1 and M2 markers [36]. At the molecular level, recent single-cell transcriptome analyses have also revealed that human microglia show multiple clusters, indicating greater heterogeneity compared to that in other mammalian species, such as mice [37]. Hence, this viewpoint presents a critical examination of the indispensability of M1/M2 macrophage activation classifications in comprehending microglial function, emphasizing their inherent constraints. Inherent in this perspective is the call for novel microglial terminology, which should be informed by various factors, including but not limited to transcriptomic and proteomic profiles, regional heterogeneity, sexual dimorphism, functions within the intact and healthy nervous system throughout the lifespan, and patterns of response to various stimuli such as physical trauma, infection, systemic inflammation, tumor, ischemia, and neurodegeneration [35,38].

## 4. Microglia as Key Players in SAE

### 4.1. Experimental Techniques

A better understanding of how microglial activation contributes to SAE may help improve its treatment (Figure 2). Animal models of sepsis are typically categorized as three types: intraperitoneal injection of lipopolysaccharide (LPS), cecal ligation perforation (CLP), and peritoneal contamination and infection (PCI). The CLP model has been widely adopted as a sepsis animal model, with well-recognized reliability and clinical relevance. The SAE model established using CLP can also cause microglial overactivation and neuronal pyroptosis, aggravating brain tissue destruction and cognitive dysfunction [39]. Another systematic review with 35 animal experiments showed that microglial activation was evident 6 h after the LPS challenge and remained for at least three days afterward [40]. However, we acknowledge that animal models of sepsis have limitations and may not reflect the high complexity of sepsis in humans. Additionally, primary microglial cells are the best candidates for microglial research. Expression profiling has led to the documentation of drastic differences between the microglia isolated immediately ex vivo and those cultured in vitro, including primary microglia and widely used cell lines such as BV-2 [41].

Historically, microglial function has primarily been studied in mouse models of disease. In order to fully understand which mouse model findings apply to humans, and whether or not microglia-targeted therapeutic approaches can be applied to human CNS disorders, it is essential to invest in innovative technologies, including human induced pluripotent stem cells (iPSCs), organoids, two-photon imaging, whole-genome transcriptomic and epigenomic analyses with complementary bioinformatics, unbiased proteomics, cytometry via time-of-flight cytometry, and complex high-content experimental models such as slice culture and zebrafish [42,43]. A systems biology approach considering multiple CNS cell types and signaling networks could provide deeper insight into SAE pathogenesis. Integrating ‘-omics’ data could help.

### 4.2. Crosstalk between Microglia, Neurons, and Astrocytes

Evidence is now accumulating that interaction between glial cells and neurons plays an active and important role in the pathophysiology of SAE. Communication between microglia and surrounding neurons is interesting; one microglial cell can come in contact with several neurons, and several microglia cells can reach one neuron [44]. When microglia become activated, they respond via chemotactic responses, migrate towards damaged neurons [45], and also release inflammatory mediators, reactive oxygen species, neurotransmitters, and other substances that can result in cytotoxic effects on neurons [46]. The CLP model was observed to cause a progressive transition of 50% of surveillant microglia towards amoeboid hypertrophic-like and gitter cell-like reactive phenotypes, characterized by active phagocytosis and frequent interaction with damaged neurons. Furthermore, microglia-mediated synaptic pruning, dependent on complement activation, was identified as a significant pathomechanism contributing to the development of neuronal abnormalities in SAE [47]. Notably, the administration of the anti-C1q complement antibody via stereotactic intrahippocampal injection was found to effectively prevent the microglial engulfment of synapses tagged with C1q [48].

Microglia also communicate with astrocytes and coordinate their responses to neuronal damage. Liddelow et al. found that a subtype of reactive astrocytes, termed A1, is induced by classically activated neuroinflammatory microglia via the secretion of Il-1α, TNF, and C1q, and that these cytokines are necessary and sufficient for the activation of A1 astrocytes [49]. A microglia–astrocyte circuit mediated by the IL-33-ST2-AKT signaling axis supports microglial metabolic adaptation and phagocytic function [50]. In the conventional perspective, it is suggested that debris from dead neurons triggers glia-mediated neuroinflammation, leading to an elevation in neuronal death. However, the expression of neurotoxic proteins in microglia alone is sufficient to trigger the death of naive neurons and propagate neuronal death through the activation of naive astrocytes toward the A1 state, and the propagation of injury is largely mediated by fragmented and dysfunctional microglial mitochondria [51]. Interferon gamma (IFNγ) remarkably increases the LPS-mediated release of TNFα and IL-1α in microglia and consequently induces the transformation of astrocytes to the A1 subtype, which ultimately results in neuronal damage. In addition, IFN-γ promotes cognitive impairment in endotoxemia by enhancing microglia-induced A1 astrocytes. Targeting IFN-γ is a novel strategy for preventing or treating cognitive dysfunction in patients with SAE [52]. The activation of the NLRP3 inflammasome in microglia induces the conversion of A1 astrocytes, thereby exacerbating a decline in neo-neurons and leading to cognitive impairment following exposure to LPS [53]. By inhibiting HMGB1/RAGE signaling, Berberine relieved sepsis-induced cognitive impairment by inhibiting microglia-stressed A1 astrocytes and neuronal decline [54]. Melatonin effectively alleviates periventricular white matter damage in septic neonatal rats, which is most likely reduced amounts of excess IL-1α, TNF-α, and C1q produced by microglia, and then the modulation of astrocyte phenotypic transformation from A1 to A2 via the MT1/JAK2/STAT3 pathway [55].

### 4.3. Microglia Activation in Cognitive Impairment

Microglial phenotypes change according to the stages and severity of the disease, which plays an essential role in SAE. Recent evidence indicates that many sepsis survivors develop long-term disabilities, including functional and cognitive impairments that affect their quality of life and ability to resume activities of daily living. Cognitive impairment is a major post-sepsis sequela that affects 12.5–21% of sepsis survivors [56] and can be progressive and permanent, although some patients may only present with transient problems. Memory, attention, and executive function are the cognitive domains most affected in sepsis survivors [57,58]. In a postmortem case–control study, microglial activation was found to be associated with delirium, and the expression of microglial markers CD68 and HLA-DR was significantly elevated in patients with delirium compared with that in controls [59]. Septic mouse models exhibited a substantial increase in chemokine production for myeloid cell recruitment, and increased neutrophil and CCR2+ inflammatory monocyte recruitment, accompanied by subtle microglial activation, which was revealed via the intravital imaging of brains [60].

Activated microglia and reactive astrogliosis, which are the hallmarks of brain injury and may contribute to synaptic deficits, were observed in septic mice [13,57,61]. IL-1β derived from activated microglia is the key molecule responsible for the hippocampal synaptic deficits observed in sepsis [57]. Most microglia were reportedly distributed around cerebral vessels 4 h after LPS injection. The extent of microglial activation was time-dependent, and the highest number of microglia was observed after 8 h in all brain regions [62]. Minocycline induced the downregulation, predominantly, of M1 markers [10], and high doses of minocycline prevented long-term potentiation impairment during sepsis [63]. Ketone body β-hydroxybutyrate (BHB) is produced in the liver and serves as an alternative energy source for the brain, heart, and skeletal muscles in mammals in states of energy deficit. The subcutaneous administration of BHB was found to enhance survival and body weight recovery in sepsis mice and improved learning and memory in sepsis-surviving mice. The improvement in learning and memory in sepsis-surviving mice was observed even when BHB was administered at the late stage of sepsis [64]. Microglial transcriptional profiling showed cholesterol that metabolism pathway genes exhibited reduced expression in males [65], and that aging microglia are unable to establish effective immune responses and sustain normal synaptic activity, directly contributing to cognitive decline [66]. As can be seen, age, sex, and genetic factors may influence microglia function during SAE.

## 5. Pharmacological Interventions Targeting Microglia

Sepsis survivorship is a prevalent and increasingly significant public health issue, characterized by significant long-term morbidity and a considerable burden of cognitive dysfunction and disability [67]. Currently, modern medicine lacks specific and effective management strategies for diagnosing and treating SAE. While early antimicrobial therapy, sufficient tissue/organ perfusion, and prompt source control during the initial stages of sepsis are recommended, there is currently no established method for preventing SAE or mitigating cognitive impairment subsequent to sepsis [68]. Therefore, there exists an urgent need for novel anti-SAE therapeutic strategies and the necessity of developing targeted treatments to mitigate the impact of SAE on the brain. Blocking microglial activation or alleviating neurotoxic reactions after microglial activation is an important therapeutic target in anti-SAE therapy, and mastering the stage-specific switching of M1/M2 phenotypes in appropriate time windows may exert therapeutic effects [69]. A summary of treatments in experimental models of SAE targeting microglia and neuroinflammation is presented in Table 1.

**Table 1 brainsci-13-01453-t001:** Treatments in experimental models of SAE targeting microglia and neuroinflammation.

References	Species, Strain, Sex	Model	Treatment and Drug Dose	Mode of Administration and Duration	Simplified Treatment Outcomes
Terrando 2010 [70]	Mouse, WT C57BL/6 and IL-1R-/-, ♂	LPS	IL-1 receptor antagonist (IL-1Ra), 100 mg/kg	Subcutaneous, immediately before LPS administration	Reduced plasma cytokine levels and hippocampal microgliosis, and ameliorated cognitive dysfunction
Li 2017 [71]	Mouse, C57 BL/6, ♂	CLP	Ginsenoside Rg1, 40 and 200 mg/kg	I.p., 1 h before the CLP operation	Improved the survival rate; suppressed IBA1 activation and learning and memory impairments
Hoshino 2017 [63]	Mouse, NA	CLP	Minocycline, 60 mg/kg	I.p., 3 consecutive days	Prevented impaired long-term potentiation in the hippocampus
Tian 2019 [72]	Mouse, C57 BL/6, ♂	LPS	Attractylon, 25 mg/kg	I.p., with LPS injection	Attenuated LPS-induced cognitive impairment, neural apoptosis, inflammatory factors, and microglial activation
Xu 2019 [73]	Mouse, BALB/c, ♂	CLP	Caspase-1 inhibitor VX765, 0.2 mg per mouse	Intragastric administration, twice daily (10 a.m. and 4 p.m.) until mice were sacrificed	Reversed cognitive dysfunction and depressive behaviors; reduced microglia activation and BBB disruption and ultrastructure damages in the brain
Michels 2019 [10]	Rat, Wistar, ♂	CLP	Minocycline, 100 μg/kg	I.c.v, immediately after CLP operation	Induced down-regulation of M1 markers
Wang 2020 [64]	Mouse, C57 BL/6, ♂	CLP	β-hydroxybutyrate, 250 mg/kg	Subcutaneous administration/i.c.v., every 6 h from the fourth day to the seventh day after CLP/twice daily for 7 days	Increased survival and body weight recovery of sepsis mice and improved learning and memory; limited neuroinflammation and neuroplasticity damage
Heimfarth 2020 [74]	Mouse, albino Swiss, ♂/♀	LPS	Indole-3-guanylhydrazone hydrochloride, 50 mg/kg	I.p., after LPS administration and for 5 consecutive days	Attenuated inflammatory reactions through the MAPK and NFκB signaling pathways, and microglia activation suppression reduced anxiety-like behavior and cognitive impairment
Xie 2020 [75]	Mouse, WT and Nrf2 KO, ♂	CLP	MCC950/Hydrogen-rich saline solution, 50 mg/kg/5 mL/kg	I.p., before operation/1 h and 6 h after CLP	Alleviated inflammation, neuronal apoptosis, and mitochondrial dysfunction via inhibiting Nrf2-mediated NLRP3 pathway.
Rocha 2021 [76]	Rat, Wistar, ♂	CLP	Anti-S100B monoclonal antibody, 10 μg/kg	I.c.v, 15 days after CLP	Increased the time of grooming; alleviated microglia activation
Bonfante 2021 [77]	Rat, Wistar, ♂	CLP	Stanniocalcin-1; 20/50/100 ng/kg	I.c.v, immediately after the CLP procedure	Improved hippocampal mitochondrial function and creatine kinase activity; reduced oxidative stress, neuroinflammation, and long-term memory impairment.
Wang 2022 [78]	Mouse, C57 BL/6, ♂	CLP	Qiang Xin 1, 0.5/1/2 g/kg	Oral, 2 h after CLP	Attenuated cognitive deficits, emotional dysfunction, and reduced neuroinflammatory responses to improve survival.
Wen 2022 [79]	Mouse, C57 BL/6 J, ♂	CLP	Cortistatin-14, 200 μg/kg	I.p., 30 min after CLP	Relieved anxiety-related behaviors and the levels of various inflammatory cytokines; reduced BBB disruption and microglial activation
Zhong 2022 [80]	Mouse, C57 BL/6, ♂	LPS	JQ-1, 50 mg/kg	I.p., 1 h before LPS	Protected the hippocampal BBB and neuronal damage and microglia activation through the attenuation of neuroinflammation
Song 2022 [81]	Mouse, C57 BL/6, ♂	LPS	Metformin, 25 mg/kg	I.p., 1 h after LPS	Blocked microglial proliferation and production of inflammatory factors
Zhong 2022 [82]	Mouse, C57 BL/J, ♂	CLP	SS-31, 5 mg/kg	I.p., once daily for 1 week	Improved the survival rate and cognitive and memory dysfunctions in CLP mice
Yang 2022 [39]	Mouse, C57 BL/6, ♂	CLP	CB2R agonist HU308, 2.5 mg/kg	I.p., three consecutive days after CLP	Inhibited microglia activity and neuronal pyroptosis
Ding 2022 [83]	Rat, NA, ♂	CLP	Fisetin, 20 mg/kg	Intragastrical administration, once a day for three consecutive days before CLP	Blocked NLRP3 inflammasome activation by promoting mitophagy and ameliorating cognitive impairment

CLP: cecal ligation and puncture; LPS: lipopolysaccharide; BBB: blood–brain barrier; NA: not announced; WT: wild type; KO: knockout; kg: kilogram; g: gram; h: hour; d: day; i.c.v.: intracerebroventricular injection; i.p.: intraperitoneal injection; male: ♂; female: ♀.

### 5.1. Blockers of Inflammatory Factors and Pyroptosis

There is a close correlation between the pathophysiology of SAE and the release of inflammatory factors and mediators. IL-1 receptor antagonist (IL-1Ra) significantly inhibited plasma cytokines, hippocampal microgliosis, and cognitive dysfunction when administered prior to symptom onset. This suggested that blocking IL-1 signaling attenuated the inflammatory cascade in response to LPS, thereby reducing microglial activation and preventing behavioral abnormalities [70]. It is believed that metformin may be able to partially reverse the severe prognosis caused by sepsis by inhibiting microglial proliferation and inflammation [81]. Cortistatin-14 is a neuropeptide structurally resembling somatostatin, which relieves anxiety-related behaviors in CLP mice, decreases the levels of various inflammatory cytokines, reduces sepsis-induced BBB disruption, and inhibits microglial activation [79]. In critically ill patients, dexmedetomidine is used as a sedative due to its ability to decrease microglial TNF-expression and alter the neuroinflammatory response of microglia [84]. Microglia are the main cells where pattern recognition receptors are expressed and pyroptosis occurs in the brain, while pyroptosis-mediated neuroinflammation is also a prominent pathogenesis of SAE [85].

In response to multiple stimuli, NLRs cleave pro-caspase-1 into activated caspase-1, and then pore-forming protein gasdermin D (GSDMD), pro-IL-1β, and pro-IL-18 were cleaved by activated caspase-1, finally leading to pyroptosis and the secretion of IL-1β [86].

The caspase-1 inhibitor VX765 inhibited caspase-1, suppressed the expression of GSDMD and its cleavage form GSDMD-NT, reduced the pyroptosis and sepsis-induced impairment of the blood–brain barrier and structural damage in the brain. Cognitive impairment and depressive symptoms were also mitigated. At days 1 and 7 after sepsis, blocking caspase-1 resulted in reduced levels of IL-1β, monocyte chemoattractant protein-1, and TNF-α in both the bloodstream and the brain [73]. As an NLRP3 inflammasome inhibitor, MCC950 was found to inhibit NLRP3 expression, IL-1β and IL-18 cytokine release, neuronal apoptosis, and mitochondrial dysfunction induced by SAE. Moreover, hydrogen inhibited the nuclear factor erythroid 2-related factor 2 (Nrf2)-mediated NLRP3 pathway, which alleviated inflammation, neuronal apoptosis, and mitochondrial dysfunction, as shown in [75]. Bromo- and extra-terminal protein inhibitor JQ1 attenuated neuroinflammation via the inhibition of the inflammasome-dependent canonical pyroptosis pathway induced via LPS injection in mice, and also selectively suppressed the activation of hippocampal microglia that protected the hippocampal BBB [80]. The inhibition of microglial activity and neuronal pyroptosis was the main mechanism by which cannabinoid type 2 receptor-specific agonist HU308 protected against SAE [39].

### 5.2. Signaling Pathway Inhibitors

Toll-like receptor 4 (TLR4) plays an essential role in promoting M1 polarization. In response to inflammatory signals, the TLR4 transmembrane receptor activates microglia and upregulates proinflammatory genes [87]. TLR4-targeting natural compounds may provide potent therapeutics for treating SAE through the TLR4/MyD88/NF-KB pathway in microglia [88]. It has also been suggested that microglial polarization is mediated by a signaling pathway linked to the mammalian target of rapamycin (mTOR) and autophagy [89], while a mTOR-mediated reduction in microglial polarization and autophagy can alleviate cerebral inflammation associated with SAE and other neurodegenerative diseases [90]. By switching microglial polarization via the mTOR-autophagy signaling pathway, hydrogen gas alleviates SAE [91].

In a study demonstrating that the IL-10 axis plays a critical role in restoring murine microglia homeostasis, neuronal impairment and fatal illness were observed in LPS-challenged mice harboring IL-10 receptor-deficient microglia [92]. Moreover, blocking IL-1β signaling ameliorated the inflammatory cascade in response to LPS, and behavioral abnormalities were reduced via microglial activation [70]. Chemokine receptor 5 (CXCR5) contributed to cognitive impairment in SAE mice via the enhancement of p38MAPK/NF-κB/STAT3 signaling, and it was found that CXCR5 knockouts restored autophagy, polarized microglia to the M2 phenotype, and inhibited the release of proinflammatory cytokines in the hippocampus by inhibiting p38MAPK and CXCR5 [93]. The aminoguanidine derivative indole-3-guanylhydrazone hydrochloride (LQM01) has been shown to have anti-inflammatory, antihypertensive, and antioxidant properties. Anxiety-like behavior and cognitive impairment induced by LPS in adult mice were reduced via LQM01 exposure during the neonatal period, which also attenuated inflammatory reactions and oxidative damage through the MAPK and NF-κB signaling pathways and microglial activation suppression [74]. Overall, these signaling pathways are critical in the progression of microglial activation through SAE, and a potential anti-SAE treatment target is blocking the above signaling pathways.

### 5.3. Mitochondrial-Targeting Drugs

In recent times, there has been a significant focus on drugs that specifically target mitochondria. An example of such a drug is the mitochondrial-targeting antioxidant peptide SS-31, which has been observed to effectively mitigate inflammation and oxidative stress in microglia stimulated with LPS through the suppression of mitochondrial fission protein 1 (Fis1) expression [94]. The administration of SS-31 among septic mice also resulted in enhanced cognitive function, increased survival rates, alleviated hippocampal inflammation, reduced reactive oxygen species production, and mitigated excessive mitochondrial fission. Moreover, SS-31 effectively decreased the activation of the NLRP3 inflammasome, inhibited the mitochondrial translocation of dynamin-related protein 1 (Drp1), reduced excessive mitochondrial fission, and attenuated the recruitment of the GSDMD N-terminal to the mitochondrial membrane in LPS-induced BV2 cell migration [82]. Recombinant human STC-1 (rhSTC1) suppressed the production of pro-inflammatory cytokines by LPS-activated microglia, and the injection of rhSTC1 into the cisterna magna reduced hippocampal inflammation and oxidative stress while increasing the activity of complex I and II of the mitochondrial respiratory chain and creatine kinase 24 h after sepsis. This was effective in preventing long-term cognitive impairment after CLP [77]. The effects of P110 are a reduction in the permeability of the BBB and the loss of tight junctions after acute LPS injury by inhibiting Drp1-Fis1 interaction [95].

Mitochondrial-targeting antioxidants may also serve as novel agents to overcome septic complications [96]. Mdivi-1, a Drp1 inhibitor, safeguarded the hippocampus against oxidative pressures and decreased the number of TUNEL-positive cells in this brain region [97]. By increasing JC-1 aggregates, antioxidant enzymes, and adenosine triphosphate levels, and decreasing ROS accumulation, Malvidin protected the cerebrum from LPS-induced mitochondrial dysfunction [98].

### 5.4. Traditional Chinese Medicine

Traditional Chinese medicine (TCM) places emphasis on the holistic treatment of individuals, while Chinese herbal medicine is distinguished by its utilization of various herbs believed to possess synergistic properties or minimize adverse effects, known as the characteristic of “Jun Chen Zuo Shi” within TCM formulations [99].

Ginsenoside Rg1 (Rg1), a prominent constituent of ginseng, exhibited enhanced postoperative survival rates and provided protection against cognitive impairments in SAE (evaluated through the Morris water maze test). Furthermore, Rg1 mitigated cerebral histopathological alterations (visualized via hematoxylin and eosin staining), suppressed IBA1 activation, and reduced the expression of inflammatory cytokines [71]. Atractylon (Atr), a prominent sesquiterpene compound found in the Asteraceae family, has demonstrated the ability to mitigate cognitive impairment, neural apoptosis, inflammatory factors, and microglial activation induced by LPS. Additionally, Atr has been observed to induce the expression of silent information regulator 1, thereby facilitating the transition of BV2 cells from a LPS-induced M1 phenotype to an M2 phenotype in vitro [72].

Qiang Xin 1 (QX1) exhibited a significant inhibition of excessive pro-inflammatory cytokine production in both peripheral and central regions, resulting in reduced microglial activation in septic mice. Furthermore, QX1 downregulated the expression of M1 phenotype microglia gene markers, such as CD32, Socs3, and CD68, while upregulating M2 phenotype marker genes, including Myc, Arg-1, and CD206 [78]. Fisetin, a constituent of the traditional Chinese medicine Cotinus coggygria, exhibits notable efficacy. Its neuroprotective properties are likely attributed to its ability to suppress neuroinflammation, as evidenced via the downregulation of IL-1 receptor, pNF-κB, TNF-α, and inducible nitric oxide synthase in microglia. Additionally, fisetin facilitates mitophagy, thereby impeding the activation of the NLRP3 inflammasome and subsequently reducing the release of IL-1β into the central nervous system. Consequently, fisetin holds promise for ameliorating cognitive impairment [83].

The above pharmacological intervention confers significant neuroprotection by inhibiting the inflammatory response in microglia and protecting against SAE, which may provide novel directions for reducing morbidity and ameliorating the neurological outcomes of SAE. Nevertheless, several limitations are important to note: microglia have beneficial housekeeping functions, and blocking microglia activation may have unintended consequences. Microglia depletion by colony-stimulating factor 1 receptor (CSF1R) inhibitor PLX3397 aggravated locomotor impairment and dopaminergic neuron loss [100]. Furthermore, the above therapies targeting microglia are in early preclinical stages, and the discussion of clinical trials on humans is limited.

## 6. Conclusions

Microglia perform several critical functions in the pathological and physiological processes of brain injury, and microglial M1 polarization is a potentially harmful mechanism in terms of neurological damage during sepsis. Modulating microglial polarization to an anti-inflammatory phenotype may serves as a potential therapeutic strategy for managing SAE to reduce morbidity and ameliorate neurological outcomes. Although several intensive studies have been conducted, the exact mechanisms and functional aspects of microglial activation remain unclear and require further exploration to reach a clear conclusion.

## Figures and Tables

**Figure 1 brainsci-13-01453-f001:**
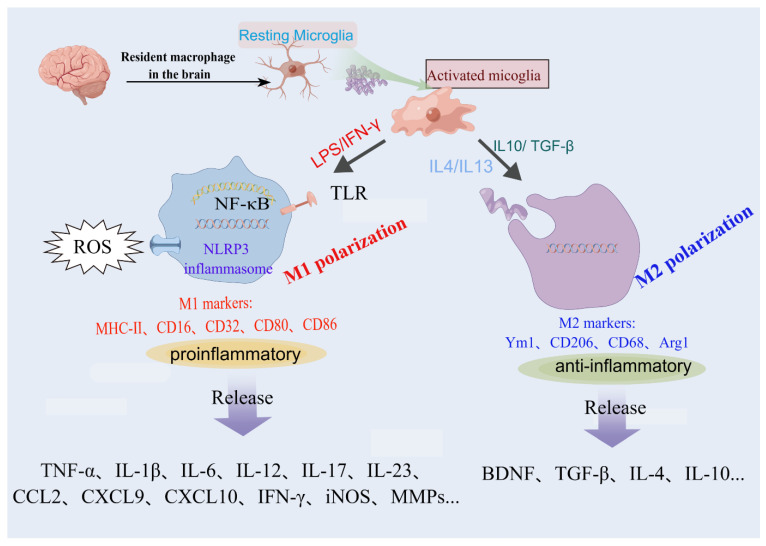
Uncontrolled or overactivated microglia are detrimental in pathological conditions. Resting microglia can be polarized to the pro-inflammatory phenotype M1 or anti-inflammatory phenotype M2 by different stimulators. This original figure was created using Figdraw (www.figdraw.com).

**Figure 2 brainsci-13-01453-f002:**
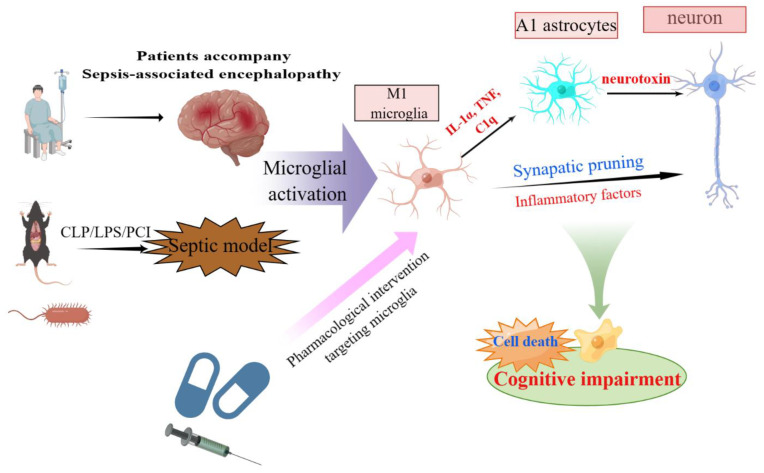
Schematic representation of functions and mechanisms of microglia-mediated neurotoxicity in SAE. Activated microglia mediate neural interactions and cell death in the CNS and play a crucial role in cognitive impairment in SAE. This original figure was created using Figdraw (www.figdraw.com). Abbreviations: SAE, sepsis-associated encephalopathy; CNS, central nervous system.

## Data Availability

Not applicable.

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
