# Peer review of "Microglial Activation: Key Players in Sepsis-Associated Encephalopathy"

_brainsci, 2023, doi:10.3390/brainsci13101453_

Round 1

Reviewer 1 Report

The manuscript is well organized and bring interesting information about SAE, microglia and how to modulate microglia in order to decrease the effects os SAE. My only concern was regarding Figure 1, as I didn't understand why the activated microglia was separated from M1 and M2 polarization.

The manuscript in general would improve with a native speaker English reviewer. Some of the sentences are very hard to understand because they are too long, such as the sentence that starts on line 70 and finishes at line 73. And the sentence that begins in line 78 for example could start with "The CNS has traditionally been considered (...)" which would make it easier to understand. Modifications like that would improve the quality of the manuscript.

Author Response

Dear reviewer:

We would like to sincerely express our great appreciation to you for your comments on our manuscript. We have studied the comments carefully and found that the comments are all valuable and very helpful for revising and improving our paper. We have carefully made corrections that we hope will be met with approval. We have submitted our document with tracked changes to highlight the revisions, and the responses to your comments are as follows:

Reviewer 1#

  1. " My only concern was regarding Figure 1, as I didn't understand why the activated microglia was separated from M1 and M2 polarization."
  2. Some of the sentences are very hard to understand because they are too long, such as the sentence that starts on line 70 and finishes at line 73. And the sentence that begins in line 78 for example could start with "The CNS has traditionally been considered (...)" which would make it easier to understand.

The author’s answer:

Reply 1:Thank you for pointing this out. Resting microglia can be activated to the pro-inflammatory phenotype M1 or alternatively to the anti-inflammatory phenotype M2 by different stimulators. We have revised the Figure 1 to address your concerns and hope that it is now clearer, the new Figure 1 reads as follows:

Reply 2:Thank you for reminding. We have added the suggested content to the manuscript (page 3, lines 79–82), the new sentence reads as follows: Microglia are derived from yolk-sac-derived progenitors and constitute approximately 10% of brain cells and approximately 20% of all glial cells, which as the primary cleaners of the brain, engage in phagocytosis to eliminate dead neurons and minimize the accumulation of debris.

We have also added the suggested content to the manuscript (page 3, lines 86–90), the new sentence reads as follows:

The CNS has traditionally been considered immune-privileged owing to the BBB tight junctions between endothelial cells, the basal lamina of these endothelial cells, and astrocytic end-feet processes, which significantly reduce the infiltration of macromolecules and immune cells to the parenchyma. In addition, the brain lacks professional antigen-presenting cells and expresses low levels of major histocompatibility complex class I and II molecules.

As suggested by the reviewer, we have carefully edited the entire manuscript and the manuscript has been polished by a professional editor before resubmission. Certificate of editing is as follows:

Thank you very much for your attention and time. Look forward to hearing from you.

Yours sincerely,

Ji-Yun Hu

Department of Critical Care Medicine, Xiangya Hospital, Central South University,

National Clinical Research Center for Geriatric Disorders, Xiangya Hospital, Central South University,

Changsha, Hunan Province 410008, P. R. China

E-mail: 218111146@csu.edu.cn

Reviewer 2 Report

The authors have endeavored to shed light on the role of microglia in SAE, which is a serious and often fatal condition frequently associated with sepsis. For the most part, the paper is well organized, but many parts of the text need rewording and some reordering, and the English language needs serious revision. There are also some inaccurate and common phrases that should be corrected.

Title – correction to lower case for sepsis

L63-65 – Please rephrase

L67-68 - And delete at the beginning

L69-70 The sentence is inappropriate, should be removed

L97-101 Unclear sentences, please rephrase

L104-106 – Too general, please rephrase more precisely

L106-108 - It should be clarified that this has only been demonstrated under experimental conditions, whereas microglia within the organism can adopt any phenotype from the spectrum between M1 and M2 phenotypes.

L108-109 - Too general a claim, reference needed.

L110-117 The entire paragraph refers to another review article - please clarify and add specific references.

L147 – “as for cell model” is concerned" – should be deleted

L151- The entire paragraph of 147/156 should be restructured

L153-156 - This paragraph should be added to line 147, before cell models. Also, lines 155 and 156 / the sentence should be restated

L162-164 – Please rephrase

L165-169 - How are mouse models related to zebrafish, etc.? This sentence does not have a clear meaning.

L171 Unclear sentence

L173 “reach a neuron” – one neuron, same neuron?

L173-174 – Please rephrase

L178-181 – Unclear sentence

L183-185 – Please rephrase

L230 – “in septic mice” – reference needed

L252-255 – please rephrase

L275-276 – please rephrase

The entire conclusions section needs to be reworded - both the English language is difficult to understand and the sentences are unclear and vague.

The English language needs extensive revision.

Author Response

Dear reviewer:

We would like to sincerely express our great appreciation to you for your comments on our manuscript. We have studied the comments carefully and found that the comments are all valuable and very helpful for revising and improving our paper. We have carefully made corrections that we hope will be met with approval. We have submitted our document with tracked changes to highlight the revisions, and the responses to your comments are as follows:

Reviewer 2#

  1. Title – correction to lower case for sepsis

L63-65 – Please rephrase

L67-68 - And delete at the beginning

L69-70 The sentence is inappropriate, should be removed OK

L97-101 Unclear sentences, please rephrase

L104-106 – Too general, please rephrase more precisely

L106-108 - It should be clarified that this has only been demonstrated under experimental conditions, whereas microglia within the organism can adopt any phenotype from the spectrum between M1 and M2 phenotypes.

L108-109 - Too general a claim, reference needed.

L110-117 The entire paragraph refers to another review article - please clarify and add specific references. 

L147 – “as for cell model” is concerned" – should be deleted

L151- The entire paragraph of 147/156 should be restructured

L153-156 - This paragraph should be added to line 147, before cell models. Also, lines 155 and 156 / the sentence should be restated

L162-164 – Please rephrase

L165-169 - How are mouse models related to zebrafish, etc.? This sentence does not have a clear meaning.

L171 Unclear sentence

L173 “reach a neuron” – one neuron, same neuron? ok

L173-174 – Please rephrase

L178-181 – Unclear sentence

L183-185 – Please rephrase

L230 – “in septic mice” – reference needed

L252-255 – please rephrase

L275-276 – please rephrase

  1. The entire conclusions section needs to be reworded - both the English language is difficult to understand and the sentences are unclear and vague.

The author’s answer:

Reply 1:Thank you for pointing this out. We have revised the text to address your concerns and hope that it is now clearer.

Title – correction to lower case for sepsis: Sepsis-associated encephalopathy (SAE) is a diffuse brain dysfunction that occurs secondary to infection in the body without overt CNS infection[1].So lower case for sepsis is applicable.

L63-65 – Please rephrase: the new sentence reads as follows: Brain development and CNS homeostasis are normally regulated by microglia processes, which include programmed cell death, clearing apoptotic newborn neurons, and pruning axons and synapses that are developing.    

L67-68 - And delete at the beginning: the new sentence reads as follows: Microglia modulate synaptic transmissions and facilitate neural circuit formation by devouring eliminated synapses in a complement-dependent manner.

L69-70 The sentence is inappropriate, should be removed OK: Inappropriate sentence has been removed.

L97-101 Unclear sentences, please rephrase: the new sentence reads as follows: Recent research has shown that microglia are most diverse in the developing, aged, and injured brain by mapping single cells of microglia in mice at various stages of development and after brain injury [2]. Depending on the milieu and factors that stimulate them, microglia can participate in classical activation, alternative activation, or acquired deactivation. Under physiological conditions, microglia are designated “resting” while the reactive morphology is termed “activated” (rounder cell body, with fewer and shorter processes or an amoeboid-shaped cell) [3].

L104-106 – Too general, please rephrase more precisely: the new sentence reads as follows: To model this change, typical experiments involve the exposure of microglial cells in vitro to stimuli such as apoptotic cells, lipopolysaccharide, inflammatory cytokines, or aggregated proteins.

L106-108 - It should be clarified that this has only been demonstrated under experimental conditions, whereas microglia within the organism can adopt any phenotype from the spectrum between M1 and M2 phenotypes: the new sentence reads as follows: Microglia can be phenotypically polarized to develop either a classical (proinflammatory, M1) or an alternative (anti-inflammatory and pro-healing, M2) phenotype (Figure 1). And it should be noticed that this has only been demonstrated under experimental conditions, whereas microglia within the organism can adopt any phenotype from the spectrum between M1 and M2 phenotypes.

L108-109 - Too general a claim, reference needed: the new sentence reads as follows: During the progression of neuroinflammatory diseases, the balance between the M1 and M2 states of microglia is dynamic. M1 microglia dominate the injury site at the end stage of the disease when immune resolution and repair processes of M2 microglia are impaired[4].

L110-117 The entire paragraph refers to another review article - please clarify and add specific references: We have clarified and added specific references [5-7] of the revised manuscript.

L147 – “as for cell model” is concerned" – should be deleted: We have deleted it.

L151- The entire paragraph of 147/156 should be restructured: the new sentence reads as follows: A better understanding of how microglial activation contributes to SAE may help improve its treatment (Figure 2). Animal models of sepsis are typically categorized as three types: intraperitoneal injection of lipopolysaccharide (LPS), cecal ligation perforation (CLP) and peritoneal contamination and infection (PCI). The CLP model has been widely adopted as a sepsis animal model, with well-recognized reliability and clinical relevance. The SAE model established using CLP can also cause microglial overactivation and neuronal pyroptosis, aggravating brain tissue destruction and cognitive dysfunction [8]. Another systematic review with 35 animal experiments showed microglial activation was evident 6 h after the LPS challenge and remained for at least three days afterward[9]. But we acknowledge that animal models of sepsis have limitations and may not reflect the high complexity of sepsis in humans. Additionally, primary microglial cells are the best candidates for microglial research. Expression profiling has documented drastic differences between the microglia isolated immediately ex vivo and those cultured in vitro, including primary microglia and widely used cell lines such as BV-2 [10].

L162-164 – Please rephrase/ L165-169 - How are mouse models related to zebrafish, etc.? This sentence does not have a clear meaning.: the new sentence reads as follows: Historically, microglial function has primarily been studied in mouse models of disease. In order to fully understand which mouse model findings apply to humans, and whether microglia-targeted therapeutic approaches can be applied to human CNS disorders, it is essential to invest in innovative technologies, including human induced pluripotent stem cells, organoids, two-photon imaging, whole-genome transcriptomic and epigenomic analyses with complementary bioinformatics, unbiased proteomics, cytometry by time-of-flight cytometry, and complex high-content experimental models such as slice culture and zebrafish [11, 12]. A systems biology approach considering multiple CNS cell types and signaling networks could provide deeper insight into SAE pathogenesis. Integrating '-omics' data could help.

L171 Unclear sentence: the new sentence reads as follows: Evidence is now accumulating that interaction between glial cells and neurons play an active and important role in the pathophysiology of SAE.

L173 “reach a neuron” – one neuron, same neuron? :the new sentence reads as follows: Communication between microglia and surrounding neurons is interesting: one microglial cell can come in contact with several neurons, and several microglia cells can reach one neuron

L173-174 – Please rephrase: the new sentence reads as follows: As soon as microglia are activated, they release inflammatory mediators, reactive oxygen species, neurotransmitters, and other substances that can result cytotoxic effects to neurons

L178-181 – Unclear sentence: the new sentence reads as follows: In addition, complement-dependent synaptic pruning by microglia as a crucial pathomechanism for the development of neuronal defects during SAE, and stereotactic intrahippocampal injection of anti-C1q complement antibody could prevent microglial engulfment of C1q-tagged synapses.

L183-185 – Please rephrase: the new sentence reads as follows: Liddelow et al. found that a subtype of reactive astrocytes, termed A1, is induced by classically activated neuroinflammatory microglia by secreting Il-1α, TNF, and C1q and that these cytokines are necessary and sufficient to activate A1 astrocytes

L230 – “in septic mice” – reference needed: we have added reference: [13-15]

L252-255 – please rephrase: the new sentence reads as follows: Therefore, there exists an urgent need for novel anti-SAE therapeutic strategies and the necessity of developing targeted treatments to mitigate the impact of SAE on the brain. Blocking microglial activation or alleviating neurotoxic reactions after microglial activation is an important therapeutic target in anti-SAE therapy, and mastering stage-specific switching of M1/M2 phenotypes in appropriate time windows may exert therapeutic effects

L275-276 – please rephrase: the new sentence reads as follows: Microglia are the main cells where pyroptosis occurs in the neurological disease, and microglial pyroptosis-mediated neuroinflammation is a prominent pathogenesis of SAE.

Reply 2:Thank you for pointing this out. We have revised the conclusion to address your concerns and hope that it is now clearer, the new sentence reads as follows: Microglia perform several critical functions in the pathological and physiological processes of brain injury, and microglial M1 polarization is a potentially harmful mechanism for neurological damage during sepsis. Modulating microglial polarization to an anti-inflammatory phenotype may serves as a potential therapeutic strategy for managing SAE to reduce morbidity and ameliorate neurological outcomes. Although several intensive studies have been conducted, the exact mechanisms and functional aspects of microglial activation remain unclear and require further exploration to reach a clear conclusion.

As suggested by the reviewer, we have carefully edited the entire manuscript and the manuscript has been polished by a professional editor before resubmission. Certificate of editing is as follows:

Thank you very much for your attention and time. Look forward to hearing from you.

Yours sincerely,

Ji-Yun Hu

Department of Critical Care Medicine, Xiangya Hospital, Central South University,

National Clinical Research Center for Geriatric Disorders, Xiangya Hospital, Central South University,

Changsha, Hunan Province 410008, P. R. China

E-mail: 218111146@csu.edu.cn

  1. Gofton TE, Young GB: Sepsis-associated encephalopathy. Nature reviews Neurology 2012, 8(10):557-566.
  2. Hammond TR, Dufort C, Dissing-Olesen L, Giera S, Young A, Wysoker A, Walker AJ, Gergits F, Segel M, Nemesh J et al: Single-Cell RNA Sequencing of Microglia throughout the Mouse Lifespan and in the Injured Brain Reveals Complex Cell-State Changes. Immunity 2019, 50(1):253-271.e256.
  3. Madry C, Kyrargyri V, Arancibia-Cárcamo IL, Jolivet R, Kohsaka S, Bryan RM, Attwell D: Microglial Ramification, Surveillance, and Interleukin-1β Release Are Regulated by the Two-Pore Domain K(+) Channel THIK-1. Neuron 2018, 97(2):299-312.e296.
  4. Hu X, Li P, Guo Y, Wang H, Leak RK, Chen S, Gao Y, Chen J: Microglia/macrophage polarization dynamics reveal novel mechanism of injury expansion after focal cerebral ischemia. Stroke 2012, 43(11):3063-3070.
  5. Fourgeaud L, Través PG, Tufail Y, Leal-Bailey H, Lew ED, Burrola PG, Callaway P, Zagórska A, Rothlin CV, Nimmerjahn A et al: TAM receptors regulate multiple features of microglial physiology. Nature 2016, 532(7598):240-244.
  6. Mizutani M, Pino PA, Saederup N, Charo IF, Ransohoff RM, Cardona AE: The fractalkine receptor but not CCR2 is present on microglia from embryonic development throughout adulthood. Journal of immunology (Baltimore, Md : 1950) 2012, 188(1):29-36.
  7. Jung S, Aliberti J, Graemmel P, Sunshine MJ, Kreutzberg GW, Sher A, Littman DR: Analysis of fractalkine receptor CX(3)CR1 function by targeted deletion and green fluorescent protein reporter gene insertion. Molecular and cellular biology 2000, 20(11):4106-4114.
  8. Yang L, Li Z, Xu Z, Zhang B, Liu A, He Q, Zheng F, Zhan J: Protective Effects of Cannabinoid Type 2 Receptor Activation Against Microglia Overactivation and Neuronal Pyroptosis in Sepsis-Associated Encephalopathy. Neuroscience 2022, 493:99-108.
  9. Hoogland IC, Houbolt C, van Westerloo DJ, van Gool WA, van de Beek D: Systemic inflammation and microglial activation: systematic review of animal experiments. J Neuroinflammation 2015, 12:114.
  10. Butovsky O, Jedrychowski MP, Moore CS, Cialic R, Lanser AJ, Gabriely G, Koeglsperger T, Dake B, Wu PM, Doykan CE et al: Identification of a unique TGF-β-dependent molecular and functional signature in microglia. Nature neuroscience 2014, 17(1):131-143.
  11. Gosselin D, Skola D, Coufal NG, Holtman IR, Schlachetzki JCM, Sajti E, Jaeger BN, O'Connor C, Fitzpatrick C, Pasillas MP et al: An environment-dependent transcriptional network specifies human microglia identity. Science (New York, NY) 2017, 356(6344).
  12. Abud EM, Ramirez RN, Martinez ES, Healy LM, Nguyen CHH, Newman SA, Yeromin AV, Scarfone VM, Marsh SE, Fimbres C et al: iPSC-Derived Human Microglia-like Cells to Study Neurological Diseases. Neuron 2017, 94(2):278-293.e279.
  13. Moraes CA, Santos G, de Sampaio e Spohr TC, D'Avila JC, Lima FR, Benjamim CF, Bozza FA, Gomes FC: Activated Microglia-Induced Deficits in Excitatory Synapses Through IL-1β: Implications for Cognitive Impairment in Sepsis. Mol Neurobiol 2015, 52(1):653-663.
  14. Hanisch UK: Microglia as a source and target of cytokines. Glia 2002, 40(2):140-155.
  15. Clarke LE, Barres BA: Emerging roles of astrocytes in neural circuit development. Nature reviews Neuroscience 2013, 14(5):311-321.

Reviewer 3 Report

The Manuscript: „ Microglial activation: key players in Sepsis-associated encephalopathy’’ by Jiyun Hu and colleagues describes the pathogenesis, morphology, function, and phenotype of microglial activation in Sepsis-associated encephalopathy and demonstrate the close relation between microglia and occurrence and development of SAE and concomitant cognitive impairment. Sepsis-associated encephalopathy (SAE) is a condition characterized by brain dysfunction that occurs as a result of sepsis, a severe systemic infection. While the exact mechanisms behind SAE are not fully understood, microglial activation is believed to be a key player in its pathogenesis. Microglia are immune cells found in the brain and central nervous system, and they play a crucial role in maintaining brain homeostasis and responding to various insults, including infections. The authors have attempted to address this point with reference to available literature.

After going through the manuscript, I have following comments for the authors:

1.     It is important to note that SAE is a complex condition influenced by multiple factors, and microglial activation is just one facet of its pathogenesis. Please mention this point in the discussion and briefly discuss other factors owing to pathogenesis in SAE.

2.     The necessity of developing targeted treatments to mitigate SAE’s effects on the brain is an important point. I would suggest the authors to elaborate this notion in the manuscript.

3.     Please mention the methodology of the review process in the concluding sentence of introduction paragraph. How were the literatures selected?

Language is fine. Minor grammatical correction and syntax adjustment needed.

Author Response

Dear reviewer:

We would like to sincerely express our great appreciation to you for your comments on our manuscript. We have studied the comments carefully and found that the comments are all valuable and very helpful for revising and improving our paper. We have carefully made corrections that we hope will be met with approval. We have submitted our document with tracked changes to highlight the revisions, and the responses to your comments are as follows:

Reviewer 3#

  1. It is important to note that SAE is a complex condition influenced by multiple factors, and microglial activation is just one facet of its pathogenesis. Please mention this point in the discussion and briefly discuss other factors owing to pathogenesis in SAE.
  2. The necessity of developing targeted treatments to mitigate SAE’s effects on the brain is an important point. I would suggest the authors to elaborate this notion in the manuscript.
  3. Please mention the methodology of the review process in the concluding sentence of introduction paragraph. How were the literatures selected?

The author’s answer:

Reply 1:Thank you for pointing this out. We have revised the text to address your concerns and hope that it is now clearer. Please see page 2-3, lines 49–53)of the revised manuscript, the new sentence reads as follows: The etiology and pathogenesis of SAE are complex, including microglial activation, blood-brain barrier (BBB) disruption, leukocyte infiltration, metabolic adaptations to systemic inflammation, bioenergetic shifts, cerebral coagulopathy or ischemia, oxidative stress due to inflammation, and mitochondrial dysfunction [1, 2].

Reply 2:Thank you for reminding. We have added the suggested content to the Discussion of the manuscript (page 7, lines 264–265), the new sentence reads as follows: Therefore, there exists an urgent need for novel anti-SAE therapeutic strategies and the necessity of developing targeted treatments to mitigate the impact of SAE on the brain.

Reply 3:Thank you for reminding. We have added the suggested content to the Discussion of the manuscript (page 3, lines 65-69), the new sentence reads as follows: In this review, we systematically searched common English databases, including PubMed, Web of Science, MEDLINE, and Embase, to investigate the critical role of microglia in SAE and summarized the prospects of therapies targeting microglial activation and neuroinflammation to alleviate cognitive impairment in SAE in recent years.

As suggested by the reviewer, we have carefully edited the entire manuscript and the manuscript has been polished by a professional editor before resubmission. Certificate of editing is as follows:

Thank you very much for your attention and time. Look forward to hearing from you.

Yours sincerely,

Ji-Yun Hu

Department of Critical Care Medicine, Xiangya Hospital, Central South University,

National Clinical Research Center for Geriatric Disorders, Xiangya Hospital, Central South University,

Changsha, Hunan Province 410008, P. R. China

E-mail: 218111146@csu.edu.cn

[1] F.A. Bozza, J.C. D'Avila, C. Ritter, R. Sonneville, T. Sharshar, F. Dal-Pizzol, Bioenergetics, mitochondrial dysfunction, and oxidative stress in the pathophysiology of septic encephalopathy, Shock 39 Suppl 1 (2013) 10-6.

[2] A.C. Bustamante, K. Opron, W.J. Ehlenbach, E.B. Larson, P.K. Crane, C.D. Keene, T.J. Standiford, B.H. Singer, Transcriptomic Profiles of Sepsis in the Human Brain, Am J Respir Crit Care Med 201(7) (2020) 861-863.

Reviewer 4 Report

This review article explores the role of microglial activation in sepsis-associated encephalopathy (SAE). Microglia are the resident immune cells of the brain that can become activated in response to infection and inflammation. In SAE, microglia become overactivated, releasing inflammatory cytokines, reactive oxygen species, and other neurotoxic factors that can damage neurons and worsen brain inflammation. Microglia can polarize towards either a pro-inflammatory M1 phenotype or an anti-inflammatory M2 phenotype depending on stimuli. In SAE, there is an imbalance towards pro-inflammatory M1 activation that drives neural damage. Patients with SAE show increased microglial activation in brain regions like the hippocampus, which may contribute to cognitive dysfunction.

Potential therapeutic approaches targeting microglia in SAE are discussed. These include blocking inflammatory cytokines like IL-1, inhibiting signaling pathways like TLR4 and NLRP3 that promote inflammation, using mitochondrial-targeted antioxidants to reduce oxidative stress, and modulating microglia towards an M2 anti-inflammatory phenotype. Treatments like minocycline, caspase-1 inhibitors, and cannabinoid receptor agonists show promise in animal models. Overall, the review highlights microglial activation as a key mechanism in SAE and discusses how targeting neuroinflammation may help prevent cognitive decline in sepsis patients. Controlling microglia overactivation and polarization may be a novel strategy for improving neurological outcomes in SAE.

  1. The review relies heavily on animal models of sepsis which may not fully replicate the complexity of human SAE. More emphasis could be placed on human studies.
  2. The characterization of M1/M2 polarization is an oversimplification. Microglia likely exist on a spectrum and take on multiple activation states.
  3. Most evidence on microglia in SAE is correlational. More functional studies are needed to prove causation between microglia activation and neuronal damage.
  4. Therapies targeting microglia are in early preclinical stages. Discussion of clinical trials in humans is limited.
  5. The review lacks quantification. Adding more meta-analysis of existing data could strengthen conclusions.
  6. Mechanisms are described but specifics are lacking. For example, which cytokines, chemokines, and transcriptional regulators are involved?
  7. There is little discussion of age, sex, and genetic factors that may influence microglia function and SAE risk.
  8. The role of peripheral immune cells and communication across the blood-brain barrier should be considered.
  9. Microglia have beneficial housekeeping functions. Blocking activation may have unintended consequences.
  10. A systems biology approach considering multiple CNS cell types and signaling networks could provide deeper insight into SAE pathogenesis. Integrating '-omics' data could help.

Author Response

Dear reviewer:

We would like to sincerely express our great appreciation to you for your comments on our manuscript. We have studied the comments carefully and found that the comments are all valuable and very helpful for revising and improving our paper. We have carefully made corrections that we hope will be met with approval. We have submitted our document with tracked changes to highlight the revisions, and the responses to your comments are as follows:

Reviewer 4#

  1. The review relies heavily on animal models of sepsis which may not fully replicate the complexity of human SAE. More emphasis could be placed on human studies.
  2. The characterization of M1/M2 polarization is an oversimplification. Microglia likely exist on a spectrum and take on multiple activation states.
  3. Most evidence on microglia in SAE is correlational. More functional studies are needed to prove causation between microglia activation and neuronal damage.
  4. Therapies targeting microglia are in early preclinical stages. Discussion of clinical trials in humans is limited.
  5. The review lacks quantification. Adding more meta-analysis of existing data could strengthen conclusions.
  6. Mechanisms are described but specifics are lacking. For example, which cytokines, chemokines, and transcriptional regulators are involved?
  7. There is little discussion of age, sex, and genetic factors that may influence microglia function and SAE risk.
  8. The role of peripheral immune cells and communication across the blood-brain barrier should be considered.
  9. Microglia have beneficial housekeeping functions. Blocking activation may have unintended consequences.
  10. A systems biology approach considering multiple CNS cell types and signaling networks could provide deeper insight into SAE pathogenesis. Integrating '-omics' data could help.

The author’s answer:

Reply 1:Thank you for pointing this out. We have revised the text to address your concerns and added some autopsy results from patients passed away with SAE. Please see page 3, lines 53–59)of the revised manuscript, the new sentence reads as follows: A localized and significant increase in CD68-positive microglia is observed in the brains of patients who die from septic shock due to severe systemic inflammation and increased microglial activity in the putamen, hippocampus, and cerebellum of the brain [1, 2]. Additionally, activation of pro-inflammatory microglia in the white matter is observed in patients with sepsis; however, this is not as evident in their gray matter. In contrast to brain inflammatory or ischemic diseases, the anti-inflammatory microglia markers CD163 and CD206 are not expressed in acute sepsis [3].

Reply 2:Thank you for reminding. We have revised the text to address your concerns (page 4-5, lines 130–144), the new sentence reads as follows: However, some scholars have different opinions on the dichotomic rigid categories of M1/M2, which is inconsistent with the wide repertoire of microglial states and functions in development, plasticity, aging, and diseases which were elucidated in recent years [4]. The advent of single-cell technologies has provided clear evidence that microglia in the living brain do not polarize to either of these phenotypes, often co-expressing M1 and M2 markers [5]. At the molecular level, recent single-cell transcriptome analyses also revealed that human microglia show multiple clusters, indicating greater heterogeneity compared to that in other mammalian species, such as mice [6]. Therefore, this perspective challenges the necessity of M1/M2 macrophage activation categories for conceptualizing microglial function, highlighting their limitations. Implicit in that opinion is the mandate for new microglial terminology, based on considerations including (but not limited to) transcriptomic and proteomic profiles, regional heterogeneity, sexual dimorphism, functions in the intact, healthy nervous system from fertilization to aging, and patterns of response to change, such as physical trauma, infection, systemic inflammation, tumor, ischemia, and neurodegeneration [4, 7].

Reply 3:Thank you for pointing this out. We have revised the text to address your concerns (page 5-6, lines 174–180), the new sentence reads as follows:

Evidence is now accumulating that interaction between glial cells and neurons play an active and important role in the pathophysiology of SAE. Communication between microglia and surrounding neurons is interesting: one microglial cell can come in contact with several neurons, and several microglia cells can reach one neuron [8]. When microglia become activated, they respond by chemotactic responses and migrate towards damaged neurons[9], and also release inflammatory mediators, reactive oxygen species, neurotransmitters, and other substances that can result cytotoxic effects to neurons [10].

Reply 4:Thank you for pointing this out. We have added this to the manuscript (page 13, lines 394–398), the new sentence reads as follows: Nevertheless, several limitations are important to note: Microglia have beneficial housekeeping functions, blocking microglia activation may have unintended consequences. Microglia depletion by colony-stimulating factor 1 receptor (CSF1R) inhibitor PLX3397 aggravated locomotor impairment and dopaminergic neuron loss [11]. And the above therapies targeting microglia are in early preclinical stages, discussion of clinical trials in humans is limited.

Reply 5:Thank you for reminding. We have added a systematic review of animal experiments [12] to the manuscript (page 5, lines 154–156), the new sentence reads as follows:

Another systematic review with 35 animal experiments showed microglial activation was evident 6 h after the LPS challenge and remained for at least three days afterward [12].

Reply 6:Thank you for reminding. We have added this to the manuscript (page 4, lines 120–129), the new sentence reads as follows: M1 microglia produce cytokines and chemokines (IL-1β, IL-6, IL-12, tumor necrosis factor α [TNF-α], and chemokine (C-C motif) ligand 2), express nicotinamide adenine dinucleotide phosphate oxidase, and generate reactive oxygen and nitrogen species. Moreover, M1 microglia express major histocompatibility complex II, CD11b, CD11c integrins, CD36, CD45, and CD47 costimulatory molecules. M2 microglia are capable of releasing several anti-inflammatory cytokines (IL-10, transforming growth factor β [TGF-β]), growth factors (insulin-like growth factor, fibroblast growth factor, and colony-stimulating factor 1), neurotrophic growth factors (nerve-derived growth factor, brain-derived neurotrophic factor, neurotrophins, and glial cell–derived neurotrophic factor)[13-15].

Reply 7:Thank you for reminding. We have revised the text to address your concerns (page 7, lines 247–251), the new sentence reads as follows: Microglial transcriptional profiling showed cholesterol metabolism pathway genes exhibited reduced expression in males [16], and aging microglia are unable to establish effective immune responses and sustain normal synaptic activity, directly contributing to cognitive decline[17]. As can be seen, age, sex, and genetic factors that may influence microglia function during SAE.

Reply 8:Thank you for reminding. We have revised the text to address your concerns (page 11-12, lines 99–101), the new sentence reads as follows: In addition to that, it has been demonstrated that microglial activation could damage BBB through release of MMP-2/-9[18].

Reply 9:Thank you for reminding. We have revised the text to address your concerns (page 11-12, lines 230–258), the new sentence reads as follows:

Nevertheless, several limitations are important to note: Microglia have beneficial housekeeping functions, blocking microglia activation may have unintended consequences. Microglia depletion by colony-stimulating factor 1 receptor (CSF1R) inhibitor PLX3397 aggravated locomotor impairment and dopaminergic neuron loss [11].

Reply 10:Thank you for reminding. We have added the text to address your concerns (page 5, lines 171–173).

As suggested by the reviewer, we have carefully edited the entire manuscript and the manuscript has been polished by a professional editor before resubmission. Certificate of editing is as follows:

Thank you very much for your attention and time. Look forward to hearing from you.

Yours sincerely,

Ji-Yun Hu

Department of Critical Care Medicine, Xiangya Hospital, Central South University,

National Clinical Research Center for Geriatric Disorders, Xiangya Hospital, Central South University,

Changsha, Hunan Province 410008, P. R. China

E-mail: 218111146@csu.edu.cn

[1] M. Michels, M.R. Abatti, P. Ávila, A. Vieira, H. Borges, C. Carvalho Junior, D. Wendhausen, J. Gasparotto, C. Tiefensee Ribeiro, J.C.F. Moreira, D.P. Gelain, F. Dal-Pizzol, Characterization and modulation of microglial phenotypes in an animal model of severe sepsis, J Cell Mol Med 24(1) (2020) 88-97.

[2] A.W. Lemstra, J.C. Groen in't Woud, J.J. Hoozemans, E.S. van Haastert, A.J. Rozemuller, P. Eikelenboom, W.A. van Gool, Microglia activation in sepsis: a case-control study, J Neuroinflammation 4 (2007) 4.

[3] T. Zrzavy, R. Höftberger, T. Berger, H. Rauschka, O. Butovsky, H. Weiner, H. Lassmann, Pro-inflammatory activation of microglia in the brain of patients with sepsis, Neuropathology and applied neurobiology 45(3) (2019) 278-290.

[4] R.C. Paolicelli, A. Sierra, B. Stevens, M.E. Tremblay, A. Aguzzi, B. Ajami, I. Amit, E. Audinat, I. Bechmann, M. Bennett, F. Bennett, A. Bessis, K. Biber, S. Bilbo, M. Blurton-Jones, E. Boddeke, D. Brites, B. Brône, G.C. Brown, O. Butovsky, M.J. Carson, B. Castellano, M. Colonna, S.A. Cowley, C. Cunningham, D. Davalos, P.L. De Jager, B. de Strooper, A. Denes, B.J.L. Eggen, U. Eyo, E. Galea, S. Garel, F. Ginhoux, C.K. Glass, O. Gokce, D. Gomez-Nicola, B. González, S. Gordon, M.B. Graeber, A.D. Greenhalgh, P. Gressens, M. Greter, D.H. Gutmann, C. Haass, M.T. Heneka, F.L. Heppner, S. Hong, D.A. Hume, S. Jung, H. Kettenmann, J. Kipnis, R. Koyama, G. Lemke, M. Lynch, A. Majewska, M. Malcangio, T. Malm, R. Mancuso, T. Masuda, M. Matteoli, B.W. McColl, V.E. Miron, A.V. Molofsky, M. Monje, E. Mracsko, A. Nadjar, J.J. Neher, U. Neniskyte, H. Neumann, M. Noda, B. Peng, F. Peri, V.H. Perry, P.G. Popovich, C. Pridans, J. Priller, M. Prinz, D. Ragozzino, R.M. Ransohoff, M.W. Salter, A. Schaefer, D.P. Schafer, M. Schwartz, M. Simons, C.J. Smith, W.J. Streit, T.L. Tay, L.H. Tsai, A. Verkhratsky, R. von Bernhardi, H. Wake, V. Wittamer, S.A. Wolf, L.J. Wu, T. Wyss-Coray, Microglia states and nomenclature: A field at its crossroads, Neuron 110(21) (2022) 3458-3483.

[5] R.M. Ransohoff, A polarizing question: do M1 and M2 microglia exist?, Nature neuroscience 19(8) (2016) 987-91.

[6] T. Masuda, R. Sankowski, O. Staszewski, C. Böttcher, L. Amann, Sagar, C. Scheiwe, S. Nessler, P. Kunz, G. van Loo, V.A. Coenen, P.C. Reinacher, A. Michel, U. Sure, R. Gold, D. Grün, J. Priller, C. Stadelmann, M. Prinz, Spatial and temporal heterogeneity of mouse and human microglia at single-cell resolution, Nature 566(7744) (2019) 388-392.

[7] K. Borst, A.A. Dumas, M. Prinz, Microglia: Immune and non-immune functions, Immunity 54(10) (2021) 2194-2208.

[8] K.L. Mueller, P.J. Hines, J. Travis, Neuroimmunology, Science (New York, N.Y.) 353(6301) (2016) 760-1.

[9] C.E. Milligan, L. Webster, E.T. Piros, C.J. Evans, T.J. Cunningham, P. Levitt, Induction of opioid receptor-mediated macrophage chemotactic activity after neonatal brain injury, Journal of immunology (Baltimore, Md. : 1950) 154(12) (1995) 6571-81.

[10] M. Prinz, T. Masuda, M.A. Wheeler, F.J. Quintana, Microglia and Central Nervous System-Associated Macrophages-From Origin to Disease Modulation, Annual review of immunology 39 (2021) 251-277.

[11] X. Yang, H. Ren, K. Wood, M. Li, S. Qiu, F.D. Shi, C. Ma, Q. Liu, Depletion of microglia augments the dopaminergic neurotoxicity of MPTP, FASEB journal : official publication of the Federation of American Societies for Experimental Biology 32(6) (2018) 3336-3345.

[12] I.C. Hoogland, C. Houbolt, D.J. van Westerloo, W.A. van Gool, D. van de Beek, Systemic inflammation and microglial activation: systematic review of animal experiments, J Neuroinflammation 12 (2015) 114.

[13] L. Fourgeaud, P.G. Través, Y. Tufail, H. Leal-Bailey, E.D. Lew, P.G. Burrola, P. Callaway, A. Zagórska, C.V. Rothlin, A. Nimmerjahn, G. Lemke, TAM receptors regulate multiple features of microglial physiology, Nature 532(7598) (2016) 240-244.

[14] M. Mizutani, P.A. Pino, N. Saederup, I.F. Charo, R.M. Ransohoff, A.E. Cardona, The fractalkine receptor but not CCR2 is present on microglia from embryonic development throughout adulthood, Journal of immunology (Baltimore, Md. : 1950) 188(1) (2012) 29-36.

[15] S. Jung, J. Aliberti, P. Graemmel, M.J. Sunshine, G.W. Kreutzberg, A. Sher, D.R. Littman, Analysis of fractalkine receptor CX(3)CR1 function by targeted deletion and green fluorescent protein reporter gene insertion, Molecular and cellular biology 20(11) (2000) 4106-14.

[16] T. Patel, T.P. Carnwath, X. Wang, M. Allen, S.J. Lincoln, L.J. Lewis-Tuffin, Z.S. Quicksall, S. Lin, F.Q. Tutor-New, C.C.G. Ho, Y. Min, K.G. Malphrus, T.T. Nguyen, E. Martin, C.A. Garcia, R.M. Alkharboosh, S. Grewal, K. Chaichana, R. Wharen, H. Guerrero-Cazares, A. Quinones-Hinojosa, N. Ertekin-Taner, Transcriptional landscape of human microglia implicates age, sex, and APOE-related immunometabolic pathway perturbations, Aging cell 21(5) (2022) e13606.

[17] J. Costa, S. Martins, P.A. Ferreira, A.M.S. Cardoso, J.R. Guedes, J. Peça, A.L. Cardoso, The old guard: Age-related changes in microglia and their consequences, Mechanisms of ageing and development 197 (2021) 111512.

[18] Z. Ruan, D. Zhang, R. Huang, W. Sun, L. Hou, J. Zhao, Q. Wang, Microglial Activation Damages Dopaminergic Neurons through MMP-2/-9-Mediated Increase of Blood-Brain Barrier Permeability in a Parkinson's Disease Mouse Model, Int J Mol Sci 23(5) (2022).

Round 2

Reviewer 2 Report

The authors have made satisfactory corrections and improved the manuscript.

 L320-321- Please rephrase the sentence:

Microglia are the main cells where pyroptosis occurs in the neurological disease CNS, and microglial pyroptosis-mediated neuroinflammation is a prominent pathogenesis of SAE [86]

The manuscript has been improved, but there are still some sentences in which either a word is missing or which need to be rephrased.

Author Response

Dear reviewer:

We would like to sincerely express our great appreciation to you for your comments on our manuscript. We have studied the comments carefully and found that the comments are all valuable and very helpful for revising and improving our paper. Thank you for pointing this out(L320-321- Please rephrase the sentence:). We have revised the text to address your concerns and hope that it is now clearer: Microglia are the main cells where pattern-recognition receptors expresses and pyroptosis occurs in the brain, pyroptosis-mediated neuroinflammation is also a prominent pathogenesis of SAE [86]

Thank you very much for your attention and time. Look forward to hearing from you.